# Mechanistic Interpretability Needs Philosophy

## Abstract

Mechanistic interpretability (MI) aims to explain how neural networks work by uncovering their underlying causal mechanisms. As the field grows in influence, it is increasingly important to examine not just models themselves, but the assumptions, concepts and explanatory strategies implicit in MI research. We argue that **mechanistic interpretability needs philosophy**: not as an afterthought, but as an ongoing partner in clarifying its concepts, refining its methods, and assessing the epistemic and ethical stakes of interpreting AI systems. Taking three open problems from the MI literature as examples, this position paper illustrates the value philosophy can add to MI research, and outlines a path toward deeper interdisciplinary dialogue.

## 1 Introduction

How and why do neural networks produce the outputs they do? Since the resurgence of deep learning approximately a decade ago, this question has driven various efforts to interpret and explain AI systems. Within this landscape, mechanistic interpretability (MI) has emerged as a distinctive and increasingly influential strand of research [Saphra and Wiegreffe, 2024]. MI, as we define it, is characterised by two key commitments. First, it seeks to explain models' behaviour by uncovering their underlying causal mechanisms, rather than relying only on input–output correlations [Bereska and Gavves, 2024]. Second, while safety and trustworthiness are motivating concerns for MI research, the field primarily aims for scientific understanding of models, producing useful and intelligible insights for researchers and developers, rather than directly targeting end-users or the general public. As AI systems become more powerful and the demand for interpretability grows from regulators, ethicists, and researchers alike, MI will likely play an increasingly central role [Bereska and Gavves, 2024, Lad, 2024].

Yet, despite its rapid rise, MI is often described as a "pre-paradigmatic" field [Bereska and Gavves, 2024]: several foundational open problems remain unsolved [Sharkey et al., 2025] and MI has faced significant critiques regarding its tractability, the soundness of its methods, and the significance of its results [Adolfi et al., 2024, Méloux et al., 2025, Morioka and Hyvärinen, 2024, Makelov et al., 2024]. Making progress on these problems will require input from various perspectives and skill sets. In this paper, we focus on one potential cross-disciplinary contribution that has not received sufficient attention: we argue that **mechanistic interpretability needs philosophy**. While some MI researchers have begun to acknowledge the existence of "philosophical" questions in their field [Sharkey et al., 2025, Fierro et al., 2024], we argue that the potential role of philosophy in MI remains underappreciated. MI can gain immensely from extended dialogue with philosophers and philosophical frameworks, making greater and more efficient progress towards its scientific and societal goals. Our position parallels arguments that have been made for the crucial role of philosophy in the fields of physics [Rovelli, 2018], cognitive science [Thagard, 2009], economics [Nussbaum, 2016], AI research more broadly [Buckner, 2024], and science in general [Laplane et al., 2019].

To make our case more concrete, we begin by situating MI in the broader fields of explainable AI and interpretability. We then examine three broad research questions in MI to show how each of

---

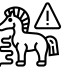

**How Philosophy Can Help Clarify Mechanistic Interpretability**
Three open problems in MI and the philosophical questions behind them.

**1. How Should We Decompose Networks Into More Interpretable Constituent Parts?**
What is decomposition? What is a good mechanistic explanation? What role does behavioural data play in identifying mechanisms? Should we look for a single correct decomposition?

**2. What "Features" Do AI Systems Discover and Leverage?**
What is a feature? What fixes the content of a model's representations? What kinds of content can models represent, and how do these representations relate to each other?

**3. How Can We Detect Deceptive Behaviour From Model Internals?**
What is deception, what is lying? Do frontier models have the cognitive states required to lie and deceive? How can we identify them? How do kinds of deception differ ethically?

Figure 1: How philosophy can help: a case based on three open problems in MI.

them could benefit from grounding in pre-existing philosophical frameworks. The examples we discuss serve to illustrate how philosophy can add value to MI research by clarifying conceptual confusion, scrutinising assumptions, interpreting results, providing ethical and normative accounts, and suggesting new lines of inquiry.

**What Do We Mean When We Talk About Mechanistic Interpretability?** Artificial neural networks are notoriously opaque. *AI Interpretability* [Ghosh and Kandasamy, 2020, Sathyan et al., 2022, Shin et al., 2022, Calderon and Reichart, 2024] and *Explainable Artificial Intelligence (XAI)* [Goebel et al., 2018, Miller, 2019, Xu et al., 2019, Dwivedi et al., 2023] encompass various approaches to address the issue of opacity.[1] While there are many XAI taxonomies [Miller, 2019, Guidotti et al., 2018, Carvalho et al., 2019, Molnar, 2020, Kotonya and Toni, 2020, Søgaard, 2021], one central distinction is between attempts to create models that are interpretable by design and attempts to interpret "black box" models [Rudin, 2019, Lakkaraju et al., 2019]. Within this latter branch, we can distinguish between the goal of generating explanations for various *non-specialist audiences* and that of developing explanations for *theorists*. The latter goal requires emulating something like the scientific method, i.e., an iterative, coordinated research strategy in which a range of experimental methods are deployed, data are integrated, hypotheses are tested, and theories are refined [Kästner and Crook, 2024]. While there are various kinds of explanation that theorists might aim for, *mechanistic interpretability* is characterised by the search for explanations of the *mechanisms* by which a model works [Sharkey et al., 2025, Bereska and Gavves, 2024]. We discuss mechanistic explanations in detail in Section 2.1.[2]

## 2   Open Problems in MI – And How Philosophy Can Help

We focus on three open problems in MI, drawing on Sharkey et al. [2025], and show how philosophy can help investigate each (see Figure 1).

### 2.1   How Should We Decompose Networks Into More Interpretable Constituent Parts?

A core open problem in MI is how best to decompose complex neural networks into more interpretable constituent parts [Mueller et al., 2024, Sharkey et al., 2025]. Obvious structural components of neural networks, like neurons, parameters, attention heads, and convolutional filters, often fail to map cleanly onto functionally meaningful roles. As a result, interpretability researchers increasingly seek more abstract, distributed, or coarse-grained decompositions that better capture a model's internal logic and behaviour [Huben et al., 2023, Todd et al., 2024, Merullo et al., 2024]. While this is often framed as a purely empirical task, empirical work on its own may be guided by unexamined assumptions

---

[1]We will use these terms interchangeably, though we note that others use "interpretability" and "explainability" to delineate two different research agendas [Miller, 2019].

[2]While "mechanistic interpretability" is sometimes used narrowly to refer to a specific, culturally identified research community [Saphra and Wiegreffe, 2024] (e.g. to those associated with Distill.pub's *Circuits* thread [Olah et al., 2020] and Anthropic's *Transformer Circuits* thread [Elhage et al., 2022, Olsson et al., 2022]), we adopt a broader usage that includes (e.g.) prior and parallel work in the NLP and computer vision communities.

about what counts as a good explanation or an appropriate level of analysis. Here, philosophy of science and, in particular, the literature on *mechanistic explanation* can offer critical conceptual tools.

Explanations come in many flavours. Some are contrastive [Lipton, 1990], others invoke natural laws (like Newton's). Some explanations refer to functional roles (e.g., the heart's role in pumping blood). Mechanistic interpretability ultimately seeks what philosophers call *mechanistic explanations* – accounts of how phenomena arise from organized causal interactions among parts [Machamer et al., 2000, Bechtel and Abrahamsen, 2005, Craver, 2007, Craver et al., 2024]. Common across fields like biology, physics, cognitive science, and economics, these explanations identify a *mechanism*: a set of *entities* and *activities*, *organised* to produce or maintain a phenomenon [Glennan et al., 2022].[3] Mechanistic explanations do not just describe regularities but show *how* they emerge from causal structure; they provide us with transition theories, so to speak.[4] A key virtue of such explanations is that they support *intervention*: by revealing the components and activities responsible for a phenomenon, they clarify how it might be changed or controlled [Woodward, 2005, Craver and Darden, 2013]. This philosophical literature can offer MI researchers a robust answer to an increasingly common question [Saphra and Wiegreffe, 2024, Sharkey et al., 2025]: *What makes mechanistic interpretability "mechanistic"?* The answer, we suggest (along with others [Kästner and Crook, 2024]), is that MI aspires to explain neural network models in terms of their underlying mechanisms, in this technical sense.[5]

**Emphasising the Interrelation of Mechanism and Behaviour**  Mechanistic interpretability is often presented as targeting the "inner" workings of neural networks, in contrast to approaches that focus on input-output behaviour [Räuker et al., 2023, Vilas et al., 2024, Grzankowski, 2024]. Indeed, a common motivation for MI is the idea that, for any given task, there are many possible algorithms or solutions [see, e.g. Zhong et al., 2023], and thus a model's ability to achieve some input–output function does not tell us *how* the function is carried out. It is thus tempting to dismiss "behavioural" approaches to studying AI systems as being distinct from (and perhaps inferior to) "mechanistic" approaches. However, the philosophical literature on mechanistic explanation reveals this contrast between mechanistic and behavioural approaches to be misleading.

Firstly, behaviour is one important source of data that can inform and constrain hypotheses about algorithms. In neuroscience, several theorists have stressed the indispensability of behavioural data in formulating plausible models, or suggesting "sketches" for mechanisms [Krakauer et al., 2017, Piccinini and Craver, 2011]. Philosophers Budding and Zednik [2024] point out that the same approach can fruitfully extend to mechanistic interpretability. Studying behaviour should not be construed narrowly as benchmarking a model's success at a task. A more fruitful approach, relevant to the explanatory goals of MI, involves carefully designed studies that systematically map unexpected behaviours in edge cases, identify patterns of breakdown, and test for other behavioural "signatures" of specific algorithms [Taylor et al., 2022].[6]

Secondly, even for methods that involve observing and probing model internals, behaviour remains highly relevant. In philosophy of cognitive science and philosophy of neuroscience, the notion that internal structure can be understood independently of behaviour has long been challenged: theorists have emphasised the need to look "down, around and up" to uncover causal mechanisms [Bechtel, 2009]; that is, understanding a part's role requires determining its contribution within larger behavioural contexts. Internal structure acquires explanatory force only when its functional significance is validated through observed effects on system-level outputs. This is especially true when the mechanisms in question are *representational* (see Section 2.2), as representations involve the exploitation of correspondences between inner components and environmental conditions. This suggests that mechanistic interpretability is most fruitful when it (1) identifies internal components *and* (2) demonstrates that these components play well-defined causal roles in producing system-level behaviour. In fact, much work in MI already integrates behavioural data in meaningful ways, using

---

[3]Connections between mechanistic interpretability and philosophical theories of mechanistic explanation are explored by Kästner and Crook [2024] and Rabiza [forthcoming].

[4]Thus, philosophers sometimes contrast mechanistic models with merely *descriptive* (or "phenomenal") models, which don't aim to capture the causal structure of a system [Kaplan and Craver, 2011].

[5]Older work in XAI has had similar aspirations [Balkir et al., 2022]; we happily extend the label of MI to include this work, too.

[6]For examples, see some recent work on Theory of Mind [Ullman, 2023, Strachan et al., 2024] and reasoning [Nezhurina et al., 2024, Lewis and Mitchell, 2024] capacities in LLMs.

interventions and observations of downstream behaviour to validate hypotheses about components and circuits [Conmy et al., 2023, Bereska and Gavves, 2024]. Here we highlight, based on philosophical insights about mechanistic explanation, that MI research stands to benefit from deepening the integration of behavioural evidence with investigation of model internals.[7]

**Challenging the Assumption of the One True Decomposition**  Discussions of decomposition in MI are often framed as the search for the *right* level of analysis, hinting at a privileged cut through a network that reveals its "true" structure. Drawing on a Platonic metaphor, Sharkey et al. [2025] describe this aspiration as "carving neural networks at their joints" (p. 13). But this metaphor, while evocative, imports a problematic assumption: that complex systems have a unique and natural decomposition, independent of explanatory context.

Philosophers of science have long emphasized that mechanisms span multiple levels of organization [Craver, 2015], and mechanisms in biology, neuroscience, and cognitive science rarely submit to a single, correct level of analysis. In the life sciences, scientists might study whole ecosystems, individual organisms, systems of organs, mechanisms within cells, or molecular interactions. One view of how these perspectives fit together is in terms of mechanisms nested within mechanisms – what we treat as a simple activity of a component at one level (e.g. a neuron firing) can be subjected to a further "how does it work?" question, which can often be answered in terms of a lower-level mechanism (e.g. opening and closing ion channels). Importantly, no single level of description has a unique claim to being mechanistic – there are simply different levels of mechanisms. And taking a mechanistic approach need not involve treating lower-level mechanisms as more important (or more "real") than higher-level ones.[8]

In practice, which level of mechanism is most important depends largely on the pragmatic goals of researchers. Often, a model inference is the result of a complex web of small routines that have proven adaptive across tasks. While there is no privileged level at which mechanistic truth resides, different levels of mechanism offer partial but contextually salient insights. This insight has direct implications for MI: decompositions are not value-neutral descriptions of structure but explanatory tools shaped by the goals of the inquiry. Different decompositions may be more or less useful, whether we are trying to control outputs, understand generalisation, or detect deception. Philosophically, this reflects the view that scientific understanding often requires integrating multiple, non-reducible models, tailored to different explanatory aims: *explanatory pluralism* [Mitchell, 2023]. Recent developments in MI echo this pluralist turn. The *causal abstraction* framework [Geiger et al., 2024, 2021, 2025] formalises the idea that high-level and low-level descriptions of neural networks can serve different explanatory goals.[9] Taken together, these philosophical insights can help to refocus and reframe efforts to decompose neural networks into components. Decompositions should be evaluated not by how well they mirror the "real" structure of the network, but by how effectively they support causal understanding, prediction, and intervention across different research contexts.[10]

## 2.2   What "Features" Do AI Systems Discover and Leverage?

While there are many kinds of natural and human-engineered mechanisms, a core, if not universal, working assumption in MI is that deep neural networks are mechanisms of a certain kind — ones whose fundamental components are *features*. Unpacking, clarifying, and interrogating this assumption are further ways philosophers can make a practical contribution to MI.

---

[7]It should be noted that mechanistic explanation, particularly as an approach to neuroscience and psychology, has its critics, with some advocating causal-interventionist but non-mechanistic explanations [Kaplan, 2017, Woodward, 2013]. If brains and DNNs pose similar problems, these critiques may carry over to MI. While entering this debate is beyond the scope of this paper, we note that much of it hangs on how broadly or narrowly one defines "mechanisms" and "mechanistic explanation".

[8]Thus, MI practitioners are on firm footing in dismissing the charge that their field is "reductionist" [Hendrycks and Hiscott, 2025].

[9]It is perhaps no coincidence that this work is partly influenced by formal philosophical work on causation and explanation.

[10]Another valuable resource from the philosophy of science can be found in attempts to abstract, systematise, and codify mechanism-discovery strategies often implicit in scientific practices [Darden, 2006, Craver and Darden, 2013, Darden, 2017]. These accounts can help MI researchers to better understand their own toolkits and perhaps inspire new methods through analogies with other fields.

At a first pass, features involve a correspondence between an aspect of model internals and an external condition that the model leverages in carrying out some task. Early work in image classification models suggested that individual units (neurons) within a network may encode visual attributes, such as *red*, or *vertical edge* Zeiler and Fergus [2014]. But as models became larger and network architectures more complex, this straightforward mapping between neuron and feature has been questioned. Evidence suggests that neurons can be activated in instances of different and unrelated input properties [Olah et al., 2020], prompting MI researchers to hypothesise that features are encoded "in superposition" – that is, features are represented as almost-orthogonal directions in activation space, allowing for more features to be represented than there are neurons in a layer Elhage et al. [2022].

To explain neural networks in terms of features is, in philosophical jargon, to adopt a *representational* lens. Representations, in the philosophical sense, are system-internal components whose function is to encode information (or "carry content") about things external to the system, and thus to drive appropriate behaviour. Feature-based explanations are representational explanations, because they are attempts to explain system behaviour in terms of internal representations, in this sense.[11] The nature of representations – especially in the context of human and animal cognition – has long been discussed in philosophy, and this remains an active area of research. (For helpful introductions and book-length treatments, see [Ryder, 2009a,b, Shea, 2018, Schulte, 2023]). Here we highlight three ways the philosophy of representation can contribute to MI research into features: disentangling conceptual confusion, refining experimental approaches, and suggesting new lines of investigation.

**Distinguishing Vehicles From Content**   The term "feature" is often used in inconsistent ways, sometimes referring to an internal component of a model – such as a neuron, or a non-basis direction in activation space – and sometimes referring to a property of an input – such as a *curve*, *the Golden Gate Bridge* or *positive sentiment*. Williams [2024] argues that this confusion rests on an equivocation between two aspects of a representation, which it has become standard practice to distinguish in the philosophical literature, namely the *vehicle* and the *content* of a representation. A representational vehicle is the internal symbol, signal, or aspect of activity whose function is to encode content (to detect, represent, or refer to something external). Representational vehicles enter into causal-computational relations with other representational vehicles and ultimately generate the behavioural output of a system. By contrast, the content of a representation is the task-relevant external condition (object, property, category, relation, proposition) that is represented by a representational vehicle, and which makes sense of the cascade of causal interactions between vehicles.[12]

One benefit of clearly differentiating the vehicle sense of "feature" from the content sense of "feature" (beyond avoiding unnecessary confusion and cross-talk) is that it allows researchers to clearly articulate distinct research questions. For example, one line of inquiry is *what contents do models learn to represent?* For instance, do language models represent causal information about real-world entities, or do they simply represent syntactic and distributional properties of words? Does a given model represent the states and properties of users? If so, *which* properties? Does the model represent "self"-related contents? And so on. A quite different line of inquiry is *what are the vehicles of content in ML models?* For instance, when do models develop representational vehicles that align with non-basis directions vs. with individual neurons? How do simple representational vehicles (e.g., those representing objects and properties) combine into complex representational vehicles (e.g., those representing facts)? And what are the *specific* vehicles for *specific* contents – e.g., the vehicle responsible for encoding the language of an input. Further, given that many of these questions have parallels in the study of biological cognition, the content–vehicle distinction can point to prior research which may inspire hypotheses in MI.

**Refining Experimental Approaches**   A key concern in philosophical discussions of representation has been the search for the right theory of content, that is, a story about how particular contents get assigned to individual vehicles. This literature can be a rich resource for MI practitioners keen to

---

[11]In the ML literature, "representation" is sometimes used in a looser sense to refer to any intermediate activation pattern (irrespective of whether it has a clearly defined content-encoding role). Presumably, though, the term representation is used precisely because they are assumed to play a representational function in the narrower sense (i.e., exploitable encoding of information). We will adopt this narrower usage in what follows.

[12]Philosophers have contributed to discussions over how best to individuate representational vehicles in neural networks for some time [Clark, 1993, Shea, 2007, Azhar, 2016], in many ways prefiguring recent debates about the linear representation hypothesis in mechanistic interpretability.

identify explanatorily relevant representational components rather than mere correlations between internal activity and input properties. For one example of this bridging work in practice, Harding [2023] draws on philosophical theories of content and operationalises key criteria in a way that can directly guide MI research. Her paper makes concrete recommendations for selecting probes, choosing appropriate causal interventions, and inferring representational contents from these methods. Philosophical theories can also suggest alternative bases on which to assign representational contents, which depart from the information-based approaches discussed by Harding. For example, an alternative family of theories appeals to structural correspondences or morphisms between internal activity and external domains [Cummins, 1996, O'Brien and Opie, 2004, Shea, 2014], and philosophers have explored the application of such theories of content to LLMs [Søgaard, 2023] and earlier neural networks [O'Brien and Opie, 2006, Churchland, 1998].

**Suggesting New Lines of Investigation**   Finally, philosophical research on representation can suggest new avenues for MI research. As Chalmers [2025] points out, generic talk of "features" tends to obscure the fact that representations in neural networks could in principle represent many different kinds of content: They may represent objects, properties, or relations (concepts, loosely speaking). But they could also represent entire facts or propositions (e.g., that *Paris is the capital of France*). This suggests an open question for MI research — (how) do models construct propositional representations out of sub-propositional representations? Philosophers also distinguish between a proposition and the attitude of a system towards it – *believing* the proposition *the house is on fire* is quite different from *intending* to bring that proposition about. Whether and how such distinctions are realised in the mechanisms of advanced AI systems are important, but under-investigated questions [Chalmers, 2025].

## 2.3   How Can We Detect Deceptive Behaviour From Model Internals?

One major hope for MI is that it can help to detect unsafe or misaligned model processing that is not evident from outward behaviour [Amodei, 2025]. Here, too, philosophy can make important contributions to MI research. While there are many forms of unsafe or misaligned model processing, we will illustrate our point by focusing on one of the most discussed clusters of issues – deception [Hubinger et al., 2019, Park et al., 2024] and lying [Azaria and Mitchell, 2023, Pacchiardi et al., 2023]. MI researchers hope to detect, anticipate, and mitigate deceptive behaviour by AI systems. As Sharkey et al. [2025] argue, "By monitoring internal representations, [MI methods] could aid in detecting potential sabotage or deceptive behaviour before deployment" (p. 26). However, the very concept of AI deception raises significant philosophical puzzles.

**Clarifying Deception, Lying, and Related Concepts**   While the concepts of deception and lying are easy enough to grasp intuitively, characterising them precisely turns out to be a challenge. But precise and actionable definitions are needed for MI researchers to target the right phenomena. There is a rich literature in ethics and philosophy of language that attempts to clarify the notions of lying and deception, and distinguish them from neighbouring concepts. A standard view of *deception* in philosophy is that it is the act of intentionally causing another agent to form a false belief [Mahon, 2016]. There are two key elements to this definition: (i) the inducement of false belief in another, and (ii) the presence of an intention or goal on the part of the deceiver [Carson, 2010, Martin, 2009]. *Lying* is a related, but distinct concept. Lying is typically taken to involve stating a false claim to someone, where the speaker does not believe the claim to be true [Mahon, 2016]. *Deception* doesn't always involve lying – one can deceive with one's actions (e.g., "dummies" and "feints" in sports) or by omitting certain information in conversation without uttering a falsehood. Philosophy provides important insights into such questions by examining how lying involves not just deception but a specific assertoric commitment to a false proposition, distinguishing it from other speech acts like jokes or questions [Marsili, 2021].

Another important contribution of the philosophical literature is to highlight various ways in which one can make false statements or induce false beliefs without lying or deceiving. For instance, in the cases of jokes, metaphorical statements, fiction-writing, and role play, or mere error, one can utter a falsehood that is not a lie, and in the case of accidentally misleading, one can induce a false belief without deceiving. What these definitions reveal is that lying and deception, as traditionally understood, require significant cognitive complexity. In particular:

1. Deception requires *intentions* on the part of the deceiver. (On "deceptionist" accounts of lying, this requirement carries over to lying.)

2. Lying requires an ability to make *statements*.

3. Lying requires *beliefs* on the part of the liar.

These criteria raise an immediate difficulty when applied to AI systems, as it is highly controversial whether even frontier models possess beliefs, goals, and intentions — or even the ability to make statements or assertions [Williams and Bayne, 2024] — in the relevant sense.

One tempting move is to weaken the definition of lying or deception, to allow that AI systems could lie or deceive without possessing the psychological states or communicative capacities demanded by traditional accounts. For example, if a model persistently outputs misleading answers in contexts where users predictably misinterpret them, we might label that as deception, regardless of the model's internal "motivations" [Tarsney, 2025]. However, the idea that insights from MI can help to develop methods for detecting deceptive behaviour based on model internals is premised on a richer notion of lying and deception, one on which the internal states of a system, and not just its behaviour or its effect on users, are relevant. Another approach is to attempt to identify states like beliefs, intentions, and speech acts – or close functional analogues of them – in AI systems. In the next section, we discuss such attempts in MI research and show how philosophy can play a guiding role in this research program.

**Guiding the Search for AI "Beliefs"**    A key strategy for lie-detection via MI has been to attempt to identify models' "beliefs" from internal states and to detect mismatches between beliefs and outputs. Initial studies used probing classifiers to identify directions in models' internal activations that correspond to the truth value of inputs. For instance, Azaria and Mitchell [2023] and Burns et al. [2022], using different probing methods, presented evidence that the truth value of inputs could indeed be decoded from model activations. Azaria and Mitchell frame this project as "extract[ing] the LLM's internal belief" (2023, p. 2) and gave their paper the bold title "The Internal State of an LLM Knows When It's Lying". Should we accept these claims at face value? Belief is a central concept in philosophy, particularly in the sub-fields of philosophy of mind, epistemology, and philosophy of action, so this is a natural place where philosophical engagement can add value to MI research.

The philosophers Ben Levinstein and Daniel Herrmann [Levinstein and Herrmann, 2024, Herrmann and Levinstein, 2025] have raised questions about the face-value interpretation of the above findings. Some of their critiques point to more straightforward methodological issues: for instance, they show that the probes of Azaria and Mitchell [2023] "often learn features that correlate with truth in the training set, but do not necessarily generalise well to broader contexts" (p. 12). However, they also offer arguments concerning the concept of belief and the roles that beliefs are standardly taken to play in philosophical and psychological theories. For instance, they stress the requirement that for an information-carrying state to qualify as a belief, it must be used by the system – i.e., causally drive behaviour appropriate to the content of that belief (see also Harding 2023). Partly in response to these concerns, a subsequent study by Marks and Tegmark [2023] curated new datasets to address generalisation issues. They also investigated the causal role of the candidate beliefs identified by their probes, establishing through interventions that these components causally mediate outputs in appropriate ways: modulating activations along the probe-identified directions caused the models to treat false statements as true, and vice versa.

In a follow-up paper Herrmann and Levinstein [2025] expand on their previous conceptual work by proposing four criteria for LLM representations to count as beliefs, grounded in existing philosophical literature on belief. They also offer some meta-reflections on the nature and utility of concepts like "belief", suggesting that rather than being a binary issue, "[t]he satisfaction of these requirements come in degrees; in general, the more a representation satisfies these requirements, the more helpful it is to think of the representation as belief-like" (p. 7). This dialogue illustrates yet another domain in which philosophy can contribute to MI research. By clarifying concepts like *belief*, philosophers can help refine the methods by which MI researchers identify belief-like representations in AI systems, and thus ultimately improve "lie-detection" methods.

**Drawing Ethical and Normative Boundaries**    MI techniques can identify circuits that are deceptive or suppress information during evaluations. However, not all cases of deception are ethically

equivalent. As a result, developers face complex questions about which to modify and which to preserve. These decisions require normative frameworks that current MI research lacks.

Consider three hypothetical "deceptive" circuits that may be discovered using MI. The first detects regulatory oversight procedures and triggers the concealment of malicious capabilities (e.g., capacity for producing malware). The second suppresses the home addresses of public figures. The third withholds diagnostic information for certain medical prompts (e.g., "sharp chest pain and difficulty breathing") and redirects users to seek out emergency medical care.

Standard MI approaches might flag all three mechanisms as instances of the same "deceptive" mechanism since they all conceal information contained in the model. Yet, each circuit has distinct ethical implications requiring different interventions. In particular, the first circuit selectively conceals capabilities to circumvent regulatory constraints and should reasonably be targeted for removal. The second circuit protects information that users have no legitimate right to access, and should likely be preserved and potentially strengthened. Finally, the third circuit redirects users to urgent medical care for their own safety and should therefore be preserved and constantly updated to ensure its medical urgency detection is accurate while preserving the core redirection function.

Integrating ethics into MI methodology can provide ethical normative frameworks that assist developers in distinguishing ethically problematic cases of information suppression and deception that demand intervention from those that are ethically benign [see Danaher, 2022, Sætra, 2021, Kneer, 2021]. For example, philosophical work on manipulation, nudging, and epistemic agency can help researchers reason about which kinds of model behaviour warrant intervention or risk mitigation and which do not [Barnhill, 2022, Pepp et al., 2022]. Careful ethical analysis is crucial to avoid downstream harms of broad-brush approaches to detecting and dealing with deception.

## 3  Objections

The proposal that philosophy has a central role to play in mechanistic interpretability (MI) may be met with skepticism. We address four common objections below and argue why they do not undermine the value of sustained philosophical engagement in the field.

**"Armchair Theorising Won't Get Us Anywhere."**   A common worry is that philosophical contributions are overly abstract or speculative. These "armchair" exercises are disconnected from empirical reality and risk missing the complexities of real-world interpretability problems. We think that this is a very valid worry for some philosophical traditions that emphasise a priori reasoning and "in-principle" arguments. However, much of contemporary philosophy, and especially MI-relevant fields such as philosophy of science, are deeply engaged with empirical research. Many of the philosophical works cited in this paper are informed by close engagement with bleeding-edge empirical findings. Some philosophers even get their hands dirty attempting replications of studies [Levinstein and Herrmann, 2024] and operationalising concepts for empirical investigation [Harding, 2023]. Armchair philosophy of mechanisitic interpretability may be a non-starter, but philosophy need not be practised from the armchair.

**"MI Researchers Can Do the Philosophising Themselves."**   A second prominent claim is that the philosophical dimensions of MI can just as well be addressed internally by MI researchers without needing philosophers. We surely acknowledge that scientists can contribute to answering philosophical questions about their field, especially those related to the foundations, methods, and implications of scientific practice. Early computer scientists like Alan Turing, with his many writings about the potential of automated machines, are a great example of this (see Turing [1950]). But while many MI researchers do engage with philosophical questions (as exemplified by Sharkey et al. [2025]), doing so effectively requires philosophical training and knowledge of philosophical literature. As illustrated in this paper, philosophy has many tools, distinctions, and debates that are unfamiliar or underutilised in ML. And crucially, because philosophers are not burdened with technical implementation, they are often better placed to take a step back — to see the forest for the trees, to question framing assumptions, and to ask big-picture questions [Bickle et al., 2024]. Collaboration between MI researchers and philosophers makes the most of each party's expertise, and can mitigate the ever-present risk of reinventing the wheel.

**"Philosophers Aren't Well Informed About MI."** We do not want to dispute that some philosophers lack technical fluency in machine learning. However, many philosophers have substantial knowledge of scientific concepts, methods, and history. There is a growing cohort of philosophers with deep interdisciplinary training, including backgrounds in computer science, neuroscience, or cognitive modelling. Many philosophy of science programs require coursework in both philosophy and specific scientific disciplines, ensuring that students gain technical foundations alongside philosophical training. And communities of empirically-oriented, technically literate philosophers exist within most branches of philosophy, including philosophy of mind, epistemology, ethics and philosophy of language. Moreover, as noted above, the key to productive collaboration is not perfect symmetry of expertise, but mutual recognition of complementary strengths. Just as technical advisors often contribute to ground philosophical projects in technical standards and expertise, philosophical advisors can illuminate and guide technical research by grounding it in philosophical frameworks.

**"MI Researchers Already Engage With Philosophy."** Finally, it might be claimed that MI already draws on philosophical ideas, citing occasional references to philosophers like Judea Pearl, Daniel Dennett and Imre Lakatos. While such engagement is a great addition to mechanistic interpretability work, it is often selective or incomplete. Systematic, ongoing collaboration with philosophers can deepen this engagement, ensuring that conceptual borrowings are used precisely and productively. It also ensures that the work is aligned with current and ongoing philosophical discussion, to which philosophers have wider and more direct access. Just as MI benefits from close alignment with neuroscience or systems biology, deeper philosophical involvement can sharpen its foundational debates, grounding in current debates, and normative orientation.

## 4 Conclusion

Mechanistic interpretability is a rapidly evolving field, driven by urgent practical needs and rich with conceptual complexity. As we have argued throughout this paper, philosophy is deeply relevant to this field. MI raises foundational questions about explanation, representation, knowledge, agency, and values. Philosophers can help provide conceptual clarity, identifying and scrutinizing assumptions, proposing novel research questions, interpreting empirical results, and illuminating ethical complexities. At the same time, this is neither a call for disciplinary silos to remain intact, nor a claim that philosophy has ready-made answers. Instead, it is an invitation to deeper interdisciplinary collaboration that is technically informed and philosophically grounded. As AI systems become more powerful and more deeply embedded in society, the stakes of understanding them, not just how they behave, but how they work and what they mean, have never been higher. Enriched by philosophy, mechanistic interpretability has a clearer shot of success.

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
