# OpenReview forum: "Mechanistic Interpretability Needs Philosophy"
_NeurIPS.cc/2025/Position_Paper_Track — Submitted to NeurIPS 2025 Position Paper Track_

### Official Review · Reviewer_Dyss · 2025-07-05

**Significance:** 2
**Presentation:** 3
**Rating:** 7
**Confidence:** 4

**Summary:**

The paper argues that progress in mechanistic interpretability is limited by conceptual ambiguities about what counts as an explanation, a feature, or even a deceptive circuit. It claims that this means philosophy can be helpful within the field, helping clarify some of these core concepts.

It breaks down mechanistic interpretability into three open problems of decomposition, feature identification, and detecting deception. In all three of these, it points out ways in which philosophy can help e.g. pointing out where one might be too narrow-minded on what constitutes a feature, the level of granularity at which we want to decompose a neural network, and what it means for a model to be deceptive.

The paper also pre-empts objections regarding whether philosophy and philosophers might be ill-equipped at contributing to mechanistic interpretability due to the technical barrier, noting both the ability for those studying the philosophy of science to study the science as well, and also the importance of some of the existing philosophical concepts and verbiage in helping talk about issues within mechanistic interpretability.

**Strengths:**

The core claim is compelling a clear: it outlines three concrete open problems within mechanistic interpretability and cleanly explains why the questions we ask and answers we seek in those problems are not answerable without considering philosophy. The paper supports this view by giving specific evidence and examples of ambiguities that are not resolved by technical means, but instead by thinking about the philosophical goals of the interpretability exercise. Since mechanistic interpretability is a growing field and a relatively pre-paradigmatic one, having a broader framework for how to build a scientific discipline around it is clearly important and relevant to NeurIPS.

**Weaknesses:**

One way in which the paper could be improved is by increasing the concreteness of the examples. While the thought experiments are compelling, it would be useful to highlight a specific case where doing philosophy has led to useful insights in mechanistic interpretability.

Another is to take more seriously critiques around whether or not the contribution of philosophy to mechanistic interpretability is necessary for making scientific progress, as opposed to merely ancillary and useful for broader understanding. The responses to the potential critiques could be expanded.

Finally, it would be useful to map some specific parts of philosophical discussion to mechanistic interpretability, since a common refrain with these cross disciplinary studies is the inability to accurately communicate across fields.

**Questions:**

1. What are some specific examples where philosophical inquiry has helped resolve conceptual confusion in mechanistic interpretability?
2. How should philosophers who are not technically versed think about contributing to mechanistic interpretability, and conversely, how should mechanistic interpretability researchers best frame their findings so philosophers of all stripes can contribute?
3. To focus in on decomposition, what is a set of philosophical heuristics which mechanistic interpretability researchers should use in order to most usefully identify the right level of decomposition?

**Alternative Position:**

Yes, and alternative positions are well-considered and named but not addressed

**Author Identification:**

No.

**Context:**

3

**Discussion:**

3

**Ethics:**

["NO or VERY MINOR ethics concerns only"]

**Position:**

Yes, the paper argues for or against a position related to machine learning.

**Support:**

3

**Thoroughness:**

4

---

### Official Review · Reviewer_yixm · 2025-08-08

**Significance:** 3
**Presentation:** 3
**Rating:** 8
**Confidence:** 4

**Summary:**

The paper argues that mechanistic interpretability can benefit greatly from cross-disciplinary contribution from philosophy. To support this claim, the authors present discussions of 3 research areas in mechanistic interpretability as examples of where MI has gained and could further gain from philosophical analysis.

**Strengths:**

The position is well argued and well grounded in the MI and philosophical literature.
If the authors' proposal were taken onboard systematically in MI practice, it would have positive transformative effects.
It effectively addresses common objections that are very likely in the minds of the MI and ML communities.

**Weaknesses:**

Granting that philosophy can be very helpful to MI, the first case study does not seem to offer the best illustration because it seems to map what is already present in MI to the corresponding literature and concepts in philosophy. This in itself is valuable, but not quite what the authors set out to demonstrate. Perhaps there are aspects of this case study that can be brought out where the contribution of philosophy has not yet been taken onboard by other means. This would make a stronger statement that interdisciplinary integration is needed a priori and not as an afterthought.

The second case study suffers, at times, from the same (minor) shortcoming.

The paragraph "Refining experimental approaches" is quite terse. It points to work and mentions the kinds of contributions that philosophy can make but does not explain any of them.

The paper is largely silent on how the cross-disciplinary interaction could/should happen in practice (with some brief exceptions in the Objections section).

As the authors point out, the call for engagement with philosophy is not novel or uncommon in areas directly adjacent to MI, such as Cognitive Science. Its value and impact will probably be confined to the MI field (no more, no less).

**Questions:**

* The paragraph on Mechanistic Interpretability makes a distinction between generating explanations for non-specialist audiences and developing explanations for theorists. It likens the latter to using the scientific method but remains silent as to how we should think of the former. How should we think about generating explanations for non-specialist audiences?
* Can case study 1 be made stronger along the lines outlined in Weakness 1?
* How do the authors envision the interdisciplinary integration could/should work in practice? Is the idea to have a philosopher on every team? Or for practitioners to keep up with the philosophical literature the way they do with the technical literature adjacent to MI.
* MI interpretability tackles many problems that have a much longer history in Cognitive Science (e.g., Adolfi et al, 2024; Comp Brain Behav). Can philosophy help identify these underlying problems early such that time and effort is not wasted in rediscovering them and their possible solutions?
* Can anything be said about how cross-disciplinary integration should happen in practice?
* Please update references to current (published) versions (e.g., Adolfi et al., 2025, ICLR; Saphra, Wiegreffe, 2024, ACL, Vilas et al., 2024, ICML)

**Alternative Position:**

Yes, and alternative positions are well-considered and addressed by the argument

**Author Identification:**

No.

**Context:**

4

**Discussion:**

4

**Ethics:**

["NO or VERY MINOR ethics concerns only"]

**Position:**

Yes, the paper argues for or against a position related to machine learning.

**Support:**

3

**Thoroughness:**

4

---

### Official Review · Reviewer_BSwe · 2025-08-11

**Significance:** 3
**Presentation:** 3
**Rating:** 8
**Confidence:** 4

**Summary:**

This paper argues that in general, interpretability would be improved as a field by working with philosophers to develop more precision in the claims made and resulting more serious thought on the questions being asked. As examples, they refer to the term mechanistic itself, the level of analysis being conducted, the meaning of the word feature, differences between belief and intent, and work on deception and lying in models that do not necessarily have beliefs.

**Strengths:**

I found this paper generally convincing and felt that many people in mechanistic interpretability could benefit from thinking more about the definitions of anthropomorphic terms that they are casually using.  I think it’s appropriate for the position track, and would not be improved by adding experiments to move it to the main track.

I like the suggestion in 103 that there should be more work on mapping unexpected behaviors in edge cases and identifying signatures of specific algorithms.

Some of the sections have really good explanations and examples that illuminate the particular concept being discussed. For example, I like those given in 214-218.

I think that the discussion of the differences between deception and lying and the behavior is documented in interpretability claiming to be out about deception and lying. Another discussion I particularly liked was about the vagueness with which we use the term feature.

I really like the mention in 273 about why a clear belief based definition of deception is specifically important for MI research in particular, because the entire premise of using MI to detect deception assumes that there are internal states which are relevant and not just external behaviors.

**Weaknesses:**

# Central thesis
I want a more specific thesis than “interpretability needs philosophy”. I didn’t see early signals as to what positions would be argued. Something like, “Mechanistic interpretability needs to be precise and interpretable. Philosophy can show how.”

# Details
Refs are glossed over, but should be explained:
- 147 explan. pluralism
- 210 Harding
- 267 statements
- 292 Azaria & Mitchell
- 298 Marks & Tegmark
- 303 Herrman & Levinstein
- 362 Bickle

# Other

84 Existing MI rarely uses causal methods, so I'm not sure it's appropriate to demand mechanisms. Paper cites other work on defining mechanistic which also disagree with this definition eg Mueller et al. (2024) and Saphra & Wiegreffe (2024)

257 Says roleplay gives untrue statements, not lies. However, they never again address roleplaying, including the defn 263-264. Yet much deception research—especially Anthropic's—has the model roleplay. When models are often understood as roleplaying, this omission is crucial.

349 "Armchair philosophy = nonstarter” Why concede? Methods change. Philosophy can predict future issues.

377: Emphasize the limitations of the small number of philosophers whom the community is familiar with.

Tense is inconsistent past/present.

**Questions:**

73: Why are functional roles not mechanistic?

226: What are the consequences or examples of relevant cases in interpretability for the conflation between belief and intent?

**Alternative Position:**

Yes, and alternative positions are well-considered and named but not addressed

**Author Identification:**

No.

**Context:**

2

**Discussion:**

3

**Ethics:**

["NO or VERY MINOR ethics concerns only"]

**Position:**

Yes, the paper argues for or against a position related to machine learning.

**Support:**

4

**Thoroughness:**

3

---

### Note · Authors · 2025-09-01

**1-10 Additional Comments:**

We thank the reviewers, meta-reviewers and organisers for considering our manuscript and giving thoughtful, useful comments. We think that the position paper stream is a welcome addition to NeurIPS and hope it continues in future iterations.

**1-11 Submit Again:**

Definitely yes

**1-1 Submission Process:**

5

**1-2 Next Year:**

No specific recommendations, but we would be very pleased to see the track continue in future iterations.

**1-4 Interest:**

["Panel discussions with other position paper authors", "Structured debates on controversial topics", "Mentorship programs for early-career researchers"]

**1-5 Thoughtful:**

8

**1-6 Supportive:**

9

**1-7 Technical Aspects Versus Position:**

9

**1-8 Gate Keeping:**

10

**1-9 Camera Ready Changes:**

To the extent that page limits allow, we plan the following main changes:

1. Clarify/unpack the central position earlier in the paper.
2. Emphasise more prominently concrete cases where philosophy has already impacted MI (e.g., Marks & Tegmark 2023 responding to Hermann & Levinstein's philosophical challenges).
3. Expand key sections: Briefly unpack citations pointed to by BSwe and flesh out "Refining Experimental Approaches" section as suggested by yixm.
4. Clarify stance on causality in definition of MI. Explain that individual MI papers needn't use causality-establishing methods, but non-causal approaches earn their place within MI by contributing to discovering causal mechanisms at the field level.
5. Follow BSwe's suggestion to highlight the methodological problem in deception research that relies on roleplay, given our argument that roleplay ≠ lying/deception.
6. Briefly discuss concrete means of increasing fruitful philosophy-MI interaction.
7. Strengthen responses to critiques, e.g. clarifying why existing engagement with commonly-cited philosophers is insufficient.
8. Technical corrections: Fix tense inconsistencies, update all references to published versions.

We are confident that these changes will strengthen our paper in line with the reviewer's comments.

**3-1 Review Response1:**

BSwe

**3-2 Reaction To Review1:**

Thank you to Reviewer BSwe for this positive and helpful review. A summary of our planned changes, including those prompted by BSwe's comments can be found above in this form. To follow up on some specific comments:

1. Thank you for pressing us to clarify the role of causality in the definition of MI. While we agree that not that every study / research paper has to include causal methods in order to count as (good) mechanistic interpretability, we maintain that that non-causal methods (e.g. correlational methods) earn their place in MI only insofar as they contribute, even if indirectly, to the project of discovering and understanding the mechanisms at play in models (which is to say structures of causally interacting components that explain system behaviours). Granted, this definition is stricter than others who say that MI encompasses anything involving analysis of model internals. But if we interpret “contribute in some way” permissively, many analyses of model internals will still qualify as MI. Thus, in practice, there might not be too much daylight between the two definitions, in terms of which work gets categorised as MI. The crucial point is that MI should be interpreted as definitionally tied to causality at the level of the goals of the field as a whole, rather than at the level of individual papers/studies.

2. Regarding functional explanations, explaining what a heart is or does by saying that it pumps blood is not (by itself) a mechanistic explanation of the heart, because it doesn’t tell you how the heart works (it doesn’t specify its components, their activities and interactions). But saying that the heart pumps blood could be (1) a way of specifying a “phenomenon” (explanandum) in need of mechanistic explanation or (2) *part* of a mechanistic explanation of a broader system (e.g. the circulatory system, the body as a whole). The functional vs. mechanistic distinction is not central to our argument, so we may drop it to avoid confusion.

**3-3 Review Response2:**

yixm

**3-4 Reaction To Review2:**

Thank you to Reviewer yixm for their encouraging review and constructive comments. A summary of our planned changes, including those prompted by yixm's comments can be found above in this form. In particular, we thank yixm for pressing us to strengthen the argument by emphasising examples of previous (and not merely potential) impacts from philosophy on MI, and to explain in detail how the cross-disciplinary interaction could/should happen in practice.

**3-5 Review Response3:**

Dyss

**3-6 Reaction To Review3:**

Thank you to Reviewer Dyss for the positive review and incisive comments. We were pleased to hear that they found the argument compelling and clear. A summary of our planned changes, including those prompted by Dyss's comments can be found above in this form. To briefly follow up on one specific comment:

On the question about whether mechanistic interpretability is necessary for making scientific progress, as opposed to merely ancillary and useful for broader understanding: we agree that our response to this worry could be strengthened. We are inclined to say that that philosophy's a contribution’s being strictly *necessary* for scientific progress (i.e. no progress could be made without it) is too high of a bar for its being of value. So long as there would be more progress with contribution from philosophy than without it, or if certain *types* of progress (e.g. on conceptual matters) are more difficult/slower without philosophy, this is a strong argument for its value in MI research. We think our paper makes a strong case for this.

---

### Meta-Review · Area_Chair_CqTv · 2025-09-12

**Rating:** 8
**Confidence:** 4

**Strengths:**

The paper presents a clear, well-argued case for why philosophy can benefit the mechanistic interpretability. It shows how the field often struggles with vague or inconsistent definitions and concepts (for example, what counts as an explanation or as a feature) and shows how philosophy can help bring more specific definitions and viewpoints.

**Weaknesses:**

The paper would benefit from concrete examples of how philosophy has already shaped mechanistic interpretability. Some section could use further refinement (for example, the section on deception could be expanded by addressing the specific challenges introduced by roleplay). In their response authors agreed to address a lot of reviewers comments.

**Questions:**

What would be the most effective way to initiate interdisciplinary dialogue that not only brings philosophers and ML researchers together but also drives real progress in mechanistic interpretability?

**Ethics:**

No, the reviewers have not raised any ethical issues.

**Thoroughness:**

2

---

### Decision · Program_Chairs · 2025-09-26

Reject